# Isotopic Transient Kinetic Analysis of Soot Oxidation on Mn_3_O_4_, Mn_3_O_4_-CeO_2_, and CeO_2_ Catalysts in Tight Contact Conditions

**DOI:** 10.3390/molecules30020343

**Published:** 2025-01-16

**Authors:** Marek Rotko, Karolina Karpińska-Wlizło

**Affiliations:** Department of Chemical Technology, Institute of Chemical Sciences, Faculty of Chemistry, Maria Curie-Skłodowska University, 3 Maria Curie-Skłodowska Square, 20-031 Lublin, Poland

**Keywords:** diesel soot oxidation, CeO_2_, Mn_3_O_4_, Mn_3_O_4_-CeO_2_, ITKA

## Abstract

The reaction mechanism of soot oxidation on Mn (Mn_3_O_4_), Mn-Ce (Mn_3_O_4_-CeO_2_), and Ce (CeO_2_) catalysts in tight contact conditions was investigated using ITKA (isotopic transient kinetic analysis). The obtained results suggest that lattice-bulk oxygen from all studied catalysts takes part in the soot oxidation process but with varying relative contributions: for the Ce catalyst, this contribution is practically 100%, whereas with decreasing Ce content in Mn-Ce catalysts, the significance of lattice-bulk oxygen for soot oxidation diminishes. For the Mn catalyst, it is estimated to be below 50%. Moreover, strong interactions between Mn and Ce ions were observed, increasing oxygen mobility in the catalyst crystal lattice and affecting the activity of Mn-Ce catalysts.

## 1. Introduction

The catalyzed diesel particulate filter (CDPF) is a crucial element of diesel exhaust purification technology aimed at reducing emissions of pollutants [1,2]. Catalytic substances enable passive regeneration of the DPF filter—the deposited soot is burned without requiring fuel to be added or heating. However, the main components of the catalytically active substances used in the CDPF filter are noble metals, in particular, platinum, which demonstrate very high catalytic activity [3,4,5]. To reduce the dependence on rare and expensive precious metals, other catalytic materials are being sought. Cerium oxide is one of the potential alternatives. Numerous studies [6,7,8,9,10] have shown its high activity in soot oxidation and its good stability. Cerium oxide can easily change its oxidation state between Ce^3+^ and Ce^4+^, which gives it an excellent ability to store and relieve high amounts of oxygen, depending on the atmosphere—oxidizing or reducing. Moreover, its catalytic properties in soot oxidation can be improved by adding a small number of various modifiers [11,12,13,14] or mixing with a large amount of various catalytically active phases of soot oxidation [15,16,17,18,19,20]. In this context, the modification of CeO_2_ through the addition of manganese oxides or mixing with them deserves special attention. Manganese oxides are among the common transition metal oxides, offering advantages such as low cost, environmental friendliness, and unique redox properties associated with their easily achievable variable valence states (Mn^2+^, Mn^3+^, and Mn^4+^). Manganese oxides demonstrate a good ability to remove soot from the DPF surface [21,22,23,24,25,26,27], especially in the presence of nitrogen oxides, because they are characterized by high catalytic activity in NO oxidation to NO_2_. NO_2_ is a well-known stronger oxidant than molecular oxygen, and technology based on this compound is commercially used to purify exhaust gases from diesel vehicles that use platinum catalysts.

In the case of the process of soot oxidation on pure cerium oxide and in tight contact conditions, it seems that the main role is played by oxygen stored inside of the crystal lattice—the Mars–van Krevelen reaction mechanism [28,29,30]. However, some researchers [31,32] propose a spillover mechanism, where the main role is played by reactive oxygen species (peroxide and superoxide ions) formed on the catalyst surface. The addition of modifiers can loosen the bonds in the crystal lattice and facilitate the migration of oxygen, but it can also cause a change in the reaction mechanism or the appearance of an alternative reaction pathway, which has been observed in previous studies [30] for the Co_3_O_4_-CeO_2_ catalyst.

According to results obtained by Wagloehner et al. [21], the reaction mechanism of soot oxidation on Mn_3_O_4_ and in tight contact conditions is similar to that observed for CeO_2_—soot is mainly oxidized by oxygen from the crystal lattice of Mn_3_O_4_. The authors estimated that the participation of bulk oxygen in soot oxidation reaches approximately 60%. It is difficult to find studies in the literature investigating the reaction mechanism of soot oxidation on pure or nearly pure Mn_3_O_4_ under tight contact conditions and in the absence of NO.

The process of soot oxidation on Mn_3_O_4_-CeO_2_ in tight contact conditions and without NO has only been studied by a few scientists [19,20]. Jampaiah et al. [19] tested catalysts with various amounts (10% to 40%) of Mn_3_O_4_ loading on CeO_2_. The highest activity was demonstrated by the catalyst containing 20% Mn_3_O_4_, and the rise in catalytic ability was assigned to the increase in redox properties. However, this study [19], as well as our previous study [20], does not supply information about the reaction mechanism. Studies on the reaction mechanism of soot oxidation on pure or nearly pure Mn_3_O_4_ under tight contact conditions and without NO are scarce in the literature. Moreover, the mechanism of soot oxidation has also been analyzed in the case of pure CeO_2_ and Mn_3_O_4_. For mechanistic studies and oxygen mobility under soot oxidation process conditions, isotopic transient kinetic analysis (ITKA) was used. The ITKA experiments were conducted according to the conditions and procedures described for steady-state isotopic transient kinetic analysis [33,34,35,36], with the soot content decreasing over time, which reduced the concentration of the product (carbon dioxide) during the experiment.

## 2. Results and Discussion

### 2.1. Catalysts’ Physicochemical Properties

The chemical composition of the prepared catalysts is presented in Table 1. The XRF results demonstrate that the obtained values of the manganese-to-cerium molar ratio are almost identical to those planned during preparation.

The results of the nitrogen adsorption/desorption experiment show that the BET surface area for manganese–cerium catalysts is more than 10 times larger compared to the manganese catalyst and close to the surface area of the cerium catalyst (Table 2). On the other hand, the average pore diameter for the manganese catalyst is several times larger, while the volume of pores is comparable to the cerium catalyst but lower than the manganese–cerium catalysts.

The crystal structure of the tested catalysts consists mainly of Mn_3_O_4_ and CeO_2_ (Table 3 and Appendix A). A small amount (about 5%) of Mn_2_O_3_ was also detected for the manganese catalyst. In the manganese–cerium catalytic systems, the crystallite size of the Mn_3_O_4_ phase is much smaller (2–9 times) than in the pure manganese catalyst. In contrast, the size of CeO_2_ crystallites is comparable to the crystallite size in the pure cerium catalyst.

The composition of the crystal structure of the catalysts was also studied using Raman spectroscopy (Figure 1). The intense band at 464 cm^−1^ is attributed to the F_2g_ vibrational mode of the cubic structure of CeO_2_ [37,38,39,40,41,42,43,44], while the second intense band at 657 cm^−1^ is assigned to the A_1g_ vibrational mode of the spinel structure of Mn_3_O_4_ [38,39,40,41,44,45,46,47]. Moreover, in the case of the pure Mn catalyst, the presence of the Mn_3_O_4_ phase is also indicated by the small bands at 288, 317, and 374 cm^−1^. On the other hand, the feeble signal observed for Mn-Ce catalysts may suggest well-mixed components, strong interaction between CeO_2_ and Mn_3_O_4_, and deformation of their structure.

The XPS results (Figure 2a,b) show that the surface of the studied catalysts consists of Mn_3_O_4_ [19,38,39,40,42,43,45,47,48,49,50] and CeO_2_ [19,38,39,40,42,43,45,48,49,50]. The detailed analysis of the 3d_3/2_ and 3d_5/2_ spectra shows that the content of Ce^3+^ ions (Appendix A) is not high and does not exceed 10%.

Moreover, the analysis of O 1s spectra suggests that different oxygen species are present on the surface of the tested catalysts (Figure 2c and Appendix A). The maximum of the signal at about 529 eV (O_α_) is attributed to lattice oxygen, while the maxima of the signal at about 531 eV (O_β_) and about 533 eV (O_γ_), according to the literature [19,38,39,40,42,43,45,48,49], correspond to various forms of absorbed oxygen (carbonates, hydroxyl groups, etc.) but can also be assigned to defects in the oxide structure.

### 2.2. Catalytic Performance

The mixed oxides demonstrate higher catalytic activity than the pure ones (Figure 3 and Appendix A). The best is the 0.25Mn-0.75Ce catalyst, followed by 0.5Mn-0.5Ce and 0.75Mn-0.25Ce. On the other hand, the worst is the Mn catalyst, followed by Ce. The difference between the most active 0.25Mn-0.75Ce catalyst and the worst is about 50 °C. All catalysts start oxidizing soot within a temperature range of 250–320 °C, while complete soot conversion is observed at around 450–500 °C.

A similar result was observed by Jampaiah et al. [19] in their study of catalysts containing varying amounts of Mn_3_O_4_ (from 10% to 40%) on CeO_2_, where the catalyst with 20% Mn_3_O_4_ showed the best performance. The high activity of this catalyst was explained by its optimal redox properties. Moreover, in the literature [51] a comparable effect can be found, among others, for Co_3_O_4_-CeO_2_ catalysts.

Besides the chemical composition, the most active catalyst (0.25Mn-0.75Ce) has similar physicochemical properties to other Mn-Ce catalysts: the bulk (Table 3 and Figure 1) and surface (Figure 2) phase composition is practically the same; the total BET surface area (Table 2) is comparable to that of the 0.5Mn-0.5Ce catalyst but slightly larger than that of the 0.75Mn-0.25Ce catalyst; and the greatest differences can be observed in the size of crystallites (Table 3). For the 0.25Mn-0.75Ce catalyst, the size of Mn_3_O_4_ crystallites is more than twice as small as that of the 0.5Mn-0.5Ce catalyst and four times smaller than that of the 0.75Mn-0.25Ce catalyst. However, caution should be exercised when considering the influence of Mn_3_O_4_ crystal size on catalytic activity, as the observed reflection intensity (Appendix A) for the Mn_3_O_4_ phase was very low. On the other hand, the size of the CeO_2_ crystallites of the 0.25Mn-0.75Ce catalyst (13.3 nm) is slightly larger than the 0.5Mn-0.5Ce catalyst (10.9 nm) and slightly smaller than the 0.75Mn-0.25Ce catalyst (17.8 nm).

### 2.3. Oxidizability and Reducibility of the Catalytic Phase

Since the calculations (Appendix A) resulting from the XPS results suggest differences in the number of oxygen vacancies and the amount of adsorbed oxygen on the surface of the Mn-Ce catalysts, their reduction and oxidation capabilities were also tested. The TPR results (Figure 4a and Appendix A) show that the Mn-Ce catalysts reduce at a lower temperature than the Mn catalyst. Furthermore, the Mn_3_O_4_ reduction steps are separated in the case of Mn-Ce catalysts. According to the literature [19,52], the first peak can be ascribed to the reduction of Mn^3+^ ions in the tetrahedral position to Mn^2+^, while the second is related to the reduction of the remaining Mn^3+^ ions to Mn^2+^. However, no significant variations in the reduction temperature were observed between the Mn-Ce catalysts, except for those due to the Mn content, with only a slight shift of the reduction maxima towards lower temperatures for the 0.25Mn-0.75Ce catalyst. CeO_2_ starts to reduce (Ce^4+^ to Ce^3+^) at temperatures above 600 °C, while the observed small peak at about 500 °C is related, according to the literature [45], to the removal of oxygen-adsorbed species. However, the TPO results (Figure 4b) suggest that the CeO_2_ reduction peak at about 500 °C can also be assigned to a partial reduction of its surface. This argument can be supported by the fact that the TPO results were obtained after the pre-reduction of the catalyst samples at 600 °C.

The TPO results (Figure 4b) show that the Mn-Ce catalysts are re-oxidized extremely easily compared to the Mn catalyst. This means that oxygen (removed during reduction at 500 °C) in the crystal lattice of the Mn-Ce catalysts can be almost completely replenished at temperatures below the temperatures of soot oxidation (Figure 3). However, one should be aware that the structure/composition of the recreated manganese oxide phase during this experiment may be completely different from the original one. Regardless, assuming that there are no structure/composition differences between the Mn-Ce catalysts, it can be stated that their various activities remain unexplained, as no shifts between the re-oxidation temperatures are seen.

### 2.4. Oxygen Mobility

The ITKA results (Figure 5a, Figure 6a, Figure 7a, Figure 8a and Figure 9a) show that oxygen from the catalyst lattice of Mn and Ce catalysts does not exchange with oxygen from the gas phase, whereas oxygen from the lattice of Mn-Ce catalysts does during soot oxidation. The number of exchanged oxygen molecules increases with increasing Mn content in the Mn-Ce catalysts. In the case of the 0.25Mn-0.75Ce catalyst, only a small amount of oxygen (Figure 8a) is involved in the exchange process, whereas for the 0.5Mn-0.5Ce (Figure 7a) and 0.75Mn-0.25Ce (Figure 6a) catalysts, these amounts are much more visible. This means that the mobility of oxygen increases with increasing Mn content in the Mn-Ce catalysts. The results suggest that oxygen exchange between the gas phase and the catalyst crystal lattice occurs independently of the soot oxidation process, as oxygen mobility changes in the opposite direction to the activity of the tested Mn-Ce catalysts. Moreover, it can also be suggested that the too-high oxygen mobility in the crystal lattice of the catalysts may not favor their catalytic abilities.

### 2.5. Reaction Mechanism

The ITKA results (Figure 5b, Figure 6b, Figure 7b, Figure 8b and Figure 9b) show that oxygen originating from the crystal lattice of Mn, Mn-Ce, and Ce catalysts takes part in the oxidation of soot. Nevertheless, the contribution of lattice-bulk oxygen to soot oxidation varies and increases with increasing Ce content in Mn-Ce catalysts. In the case of the Mn catalyst (Figure 5b), the concentration of C^16^O_2_ drops to about 20% within 100–150 s. After this time, the rate of C^16^O_2_ decline slows down drastically. At the same time, the C^18^O_2_ signal demonstrates similar but opposite changes. The C^16^O^18^O signal initially increases at a high rate, reaching a maximum after 10–20 s, and later, its values decline to a more or less stable position. The C^16^O_2_, C^16^O^18^O, and C^18^O_2_ signals demonstrate this after switching between isotopic mixtures, the concentration of CO_2_ with the old oxygen atom(s) (^16^O) decreases to approximately 40%. The behavior of the C^18^O_2_ signal just after isotopic switching indicates that soot is oxidized parallelly by oxygen from the crystal lattice bulk of the Mn catalyst and by surface oxygen. This means that the soot oxidation process takes place according to two competing/parallel reaction pathways/mechanisms: the first one is the classical redox mechanism (with lattice-bulk oxygen), while the second one is related to the oxidation of soot with oxygen chemically adsorbed on the catalyst surface and/or lattice oxygen only from the surface. These conclusions are consistent with the results obtained by Wagloehner et al. [21], which suggest that the participation of bulk oxygen in soot oxidation reaches approximately 60%. Moreover, the obtained results for the Mn catalyst are very similar to our previous results [30] obtained for Co_3_O_4_. However, pure Co_3_O_4_ demonstrates slightly lower participation of bulk oxygen in soot oxidation than Mn_3_O_4_.

In the case of the Ce catalyst, the ITKA results (Figure 9b) are almost identical to those for the CeO_2_ catalyst studied in our previous paper [30]. These results suggest the Mars–van Krevelen reaction mechanism because the observed C^18^O_2_ signal after the isotopic switch increases very slowly (after 50 s or even 100 s, it is still practically invisible). Similar behavior can be observed for the C^16^O^18^O signal—it also slowly increases, while the C^16^O_2_ signal slowly decreases. Of course, these results cannot completely exclude the participation of chemisorbed oxygen and/or lattice oxygen only from the surface in the soot oxidation process, but its influence may only be very small.

In the case of the Mn-Ce catalysts, the observed (Figure 6b, Figure 7b, Figure 8b) delays between the C^16^O_2_ curve and the Kr curve are smaller than for the Ce catalyst but still larger than for the Mn catalyst. Furthermore, the growth of the C^18^O_2_ concentration is quicker compared to the Ce catalyst and slightly gentler than for the Mn catalyst. The signal of C^16^O^18^O grows much faster than for the Ce catalyst but still slower than for the Mn catalyst. This means that the mechanism of soot oxidation on the surface of the Mn-Ce catalysts differs slightly from those observed on the surface of pure CeO_2_ and Mn_3_O_4_. However, it can generally be stated that the reaction mechanism proposed for Mn-Ce catalysts must partially contain elements characteristic of both CeO_2_ and Mn_3_O_4_ since the observed shape of the C^16^O_2_, C^16^O^18^O, and C^18^O_2_ signals is intermediate/between the signals typical of CeO_2_ and Mn_3_O_4_. Additionally, the C^16^O_2_, C^16^O^18^O, and C^18^O_2_ signals observed for Mn-Ce catalysts depend on the element content. Increasing the Ce content in Mn-Ce catalysts causes a larger number of oxygen atoms from the deeper layers of the catalyst lattice to participate in soot oxidation. In the case of the 0.25Mn-0.75Ce, the process of soot oxidation proceeds according to the Mars–van Krevelen mechanism, similar to pure CeO_2_, and the probability of soot oxidation according to the alternative mechanism(s) can be limited to a few percent. On the other hand, both 0.75Mn-0.25Ce and 0.5Mn-0.5Ce catalysts demonstrate a much greater impact of chemisorbed oxygen and/or lattice oxygen only from the surface on the process of soot oxidation, making their behavior closer to that of pure Mn_3_O_4_.

Summing up this part, it can be stated that the interaction between CeO_2_ and Mn_3_O_4_ increases the catalyst redox properties of catalysts (accelerating the transfer of oxygen within the crystal lattice), and consequently, the Mn-Ce catalysts are more active than pure CeO_2_. However, as the amount of Mn_3_O_4_ increases, the number of active centers generated by CeO_2_ decreases, resulting in the lower activity of 0.75Mn-0.25Ce and 0.5Mn-0.5Ce than 0.25Mn-0.75Ce.

## 3. Materials and Methods

### 3.1. Catalyst Preparation

Catalytic materials were prepared with the co-precipitation method using aqueous solutions of cerium and manganese acetate (Sigma-Aldrich, Burlington, MA, USA). Synthesis was carried out in a jacketed reactor (Radleys, Saffron Walden, UK) at 60 °C by dosing (5 cm^3^/min) the precipitation agent (ammonium carbonate) with a peristaltic pump until pH = 7.8. The obtained suspension was stirred using a mechanical mixer at 300 rpm. After two hours of conditioning, the precipitate was filtered off, washed with deionized water to pH = 7, and pre-dried with anhydrous alcohol. The filtration, washing, and pre-drying operations were carried out using a centrifuge. The catalyst precursors were dried at 110 °C and calcinated at 500 °C for 2 h. For testing in the catalytic reactor (activity and ITKA measurements), the catalyst samples were mixed with soot (Printex U, Evonik Industries, Essen, Germany), pressed, and then ground to a fraction of 0.15–0.3 mm to obtain a tight contact mixture.

### 3.2. Physicochemical Characteristics

The metal content (Mn and Ce) of the tested catalytic materials was obtained via X-ray fluorescence (XRF). The total BET surface area, average pore size, and volume of pores were calculated from the physical nitrogen adsorption/desorption results. Using the XRD (X-ray diffraction) method, the type and share of crystallographic phases and the mean size of their crystallites were determined—more details are available in a previous publication [37].

The solid-phase composition of the tested catalytic materials was also analyzed via Raman spectroscopy using an inVia Reflex Renishaw spectrometer (Renishaw plc, Wotton-under-Edge, UK). The spectra were collected in the range from 100 cm^−1^ to 1000 cm^−1^ with a resolution of 1 cm^−1^, after exposure to a 785 nm laser.

The surface of the tested catalytic materials was examined via X-ray photoelectron spectroscopy (XPS). For these purposes, a Prevac system (Rogów, Poland) with a monochromatic Al Kα source and a hemispherical electron analyzer, both manufactured by VG Scienta (Hastings, UK), was used. The studies were performed at an analyzer energy of 200 eV and an energy step of 0.5 eV and 0.1 eV for survey scan and high-resolution spectra, respectively.

### 3.3. Catalytic Studies

The activity of the catalytic materials was examined in the Microactivity Reference system from PID Eng& Tech (Madrid, Spain). The catalyst samples during the activity test contained 50 mg of the catalyst–soot mixture diluted with 450 mg of sand. The reaction mixture stream fed to the reactor at a rate of 100 cm^3^/min consisted of 10 vol.% of oxygen, 10 vol.% of argon (internal standard), and helium as balance gas. The gas composition of the post-reaction mixture at the outlet of the fixed-bed quartz reactor was analyzed using an HPR 20 R&D quadrupole mass spectrometer from Hiden Analytical (Warrington, UK). With the progression of the analysis, the temperature of the catalytic reactor was increased linearly at the rate of 5 °C/min until all amounts of soot were oxidized.

### 3.4. Temperature-Programmed Reduction/Oxidation

Temperature-programmed reduction (TPR) and oxidation (TPO) were performed in an AutoChem II 2920 system (Micromeritics, Norcross, GA, USA) by heating the catalyst samples from 45 °C to 900 °C with a ramp of 10 °C/min. First, 50 mg of catalyst samples and mixtures of 5% O_2_ in He and 5% H_2_ in Ar fed at the rate of 30 cm^3^/min were used. Before the TPR experiment, the catalytic samples were heated in O_2_/He to the calcination temperature, while before the TPO experiment, the catalytic samples were reduced in H_2_/Ar at 600 °C for 2 h.

### 3.5. Isotopic Transient Kinetic Analysis

The ITKA experiments were conducted in an apparatus similar to that used for the activity studies. Moreover, the composition of the catalyst bed and the total volumetric flow rate of the reaction mixture were the same. The reaction mixture had a slightly different composition and contained 10 vol.% of oxygen (^16^O_2_ or ^18^O_2_), 10 vol.% of krypton (inert tracer), 10 vol.% of argon (internal standard), and helium as balance gas. The switching between ^16^O_2_/Kr/Ar/He and ^18^O_2_/Ar/He took place at a temperature corresponding to 40% soot conversion and after 10 min under isothermal conditions.

## 4. Conclusions

The results of our catalytic research confirm that the Mn-Ce catalysts (Mn_3_O_4_-CeO_2_) demonstrate higher activity than pure oxides (Mn_3_O_4_ and CeO_2_) in soot oxidation under tight contact conditions. Moreover, the activity of Mn-Ce catalysts can be arranged in the following sequence: 0.25Mn-0.75Ce > 0.5Mn-0.5Ce > 0.75Mn-0.25Ce. The most active catalyst (0.25Mn-0.75Ce) has, among others, the largest total surface area and the smallest Mn_3_O_4_ crystallite size and shows the lowest oxygen mobility. However, the reaction mechanism of soot oxidation on the 0.25Mn-0.75Ce catalyst is the strongest, similar to the pure CeO_2_. The main amount (practically all) of soot is oxidized using oxygen from the catalyst crystal lattice bulk (Mars–van Krevelen mechanism). In the case of the remaining Mn-Ce catalysts (0.5Mn-0.5Ce and 0.75Mn-0.25Ce), the impact of chemisorbed oxygen and/or lattice oxygen only from the surface on soot oxidation is higher and increases with increasing Mn_3_O_4_ content. The obtained results for pure Mn_3_O_4_ indicate that soot oxidation occurs simultaneously via two different mechanisms: one involving chemisorbed oxygen and/or lattice oxygen from the surface and the other involving lattice-bulk oxygen, with both pathways likely having a similar quantitative influence.

## Figures and Tables

**Figure 1 molecules-30-00343-f001:**
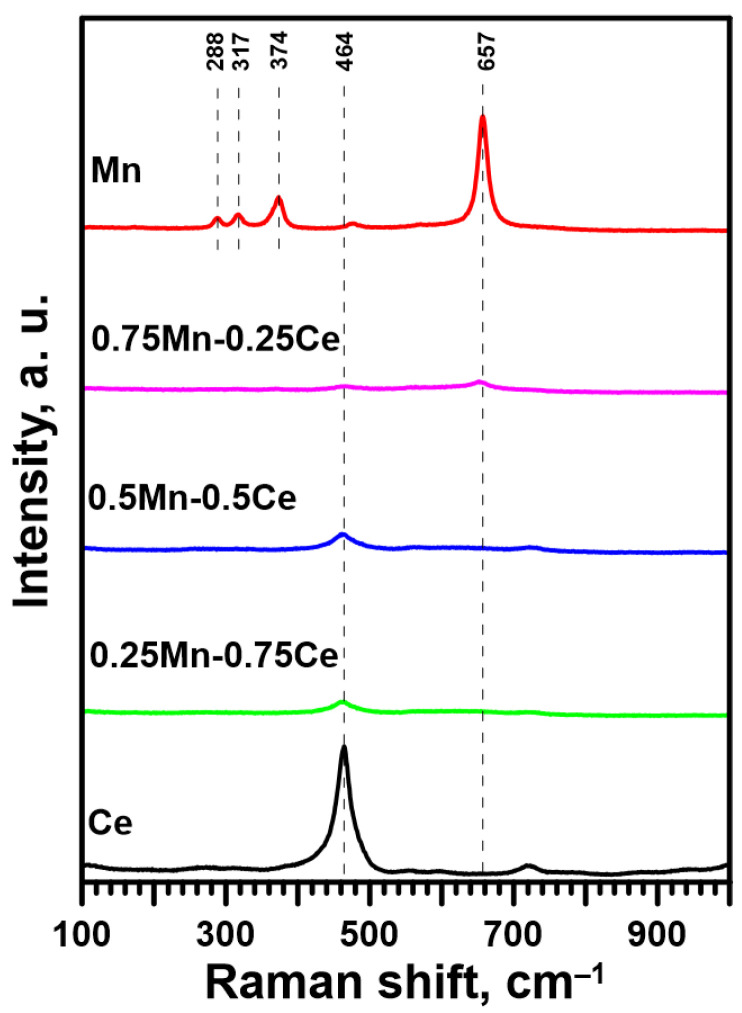
Raman spectra for Mn, Mn-Ce, and Ce catalysts.

**Figure 2 molecules-30-00343-f002:**
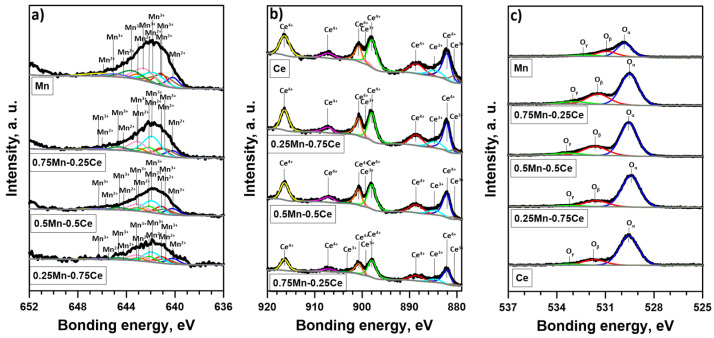
High-resolution XPS spectra of Mn 2p_3/2_ (**a**), Ce 3d_3/2_ and Ce 3d_5/2_ (**b**), and O 1s (**c**) obtained for the tested catalysts.

**Figure 3 molecules-30-00343-f003:**
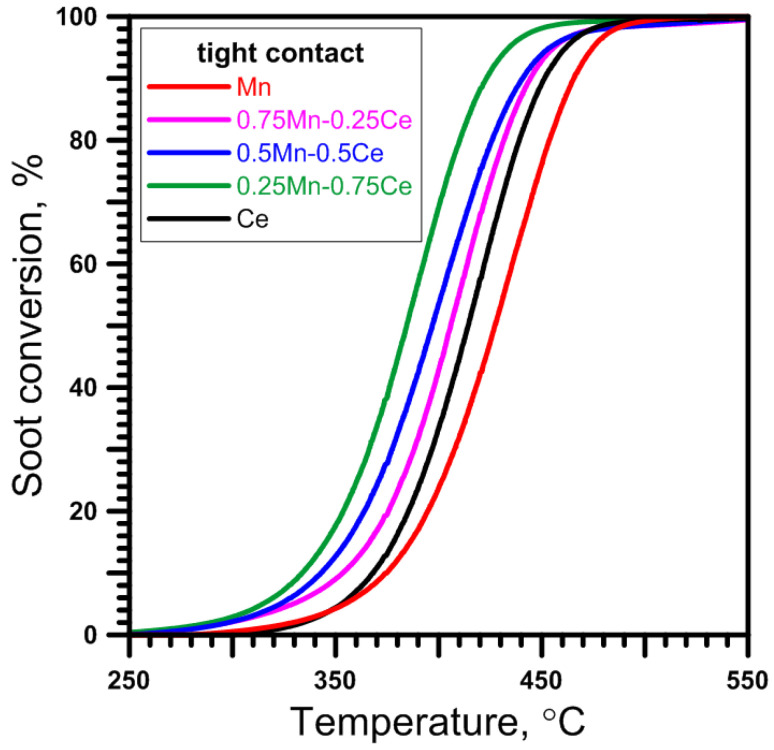
Activity of Mn, Mn-Ce, and Ce catalysts in soot oxidation under tight contact conditions.

**Figure 4 molecules-30-00343-f004:**
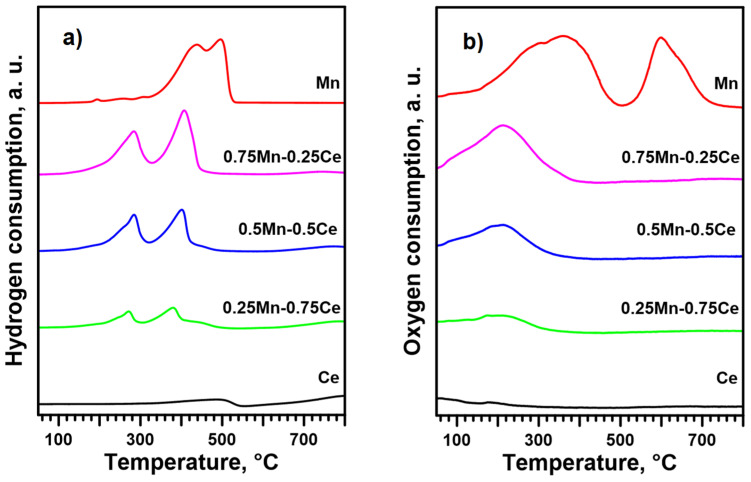
TPR (**a**) and TPO (**b**) results for the analyzed catalysts.

**Figure 5 molecules-30-00343-f005:**
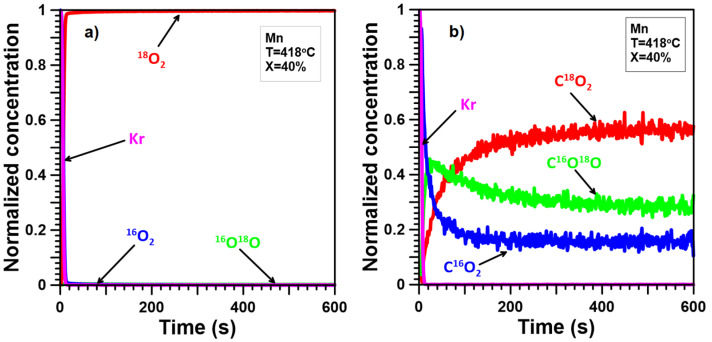
ITKA results obtained for the Mn catalyst at the temperature corresponding to 40% soot conversion ((**a**)—changes in ^16^O_2_, ^16^O^18^O, and ^18^O_2_ concentration; (**b**)—changes in C^16^O_2_, C^16^O^18^O, and C^18^O_2_ concentration observed after switching from ^16^O_2_/Kr/Ar/He to ^18^O_2_/Ar/He).

**Figure 6 molecules-30-00343-f006:**
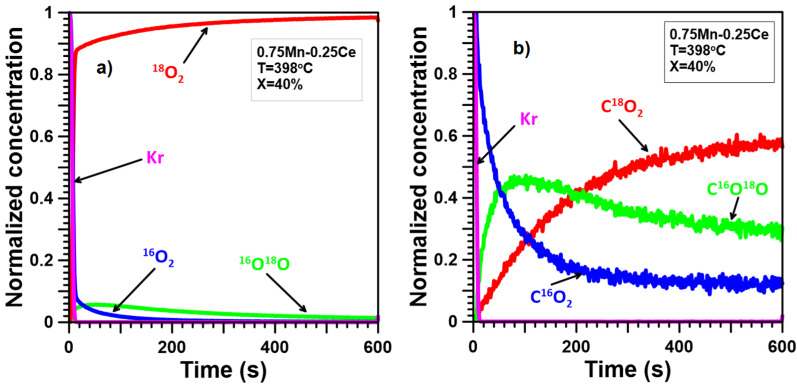
ITKA results obtained for the 0.75Mn-0.25Ce catalyst at the temperature corresponding to 40% soot conversion ((**a**)—changes in ^16^O_2_, ^16^O^18^O, and ^18^O_2_ concentration; (**b**)—changes in C^16^O_2_, C^16^O^18^O, and C^18^O_2_ concentration observed after switching from ^16^O_2_/Kr/Ar/He to ^18^O_2_/Ar/He).

**Figure 7 molecules-30-00343-f007:**
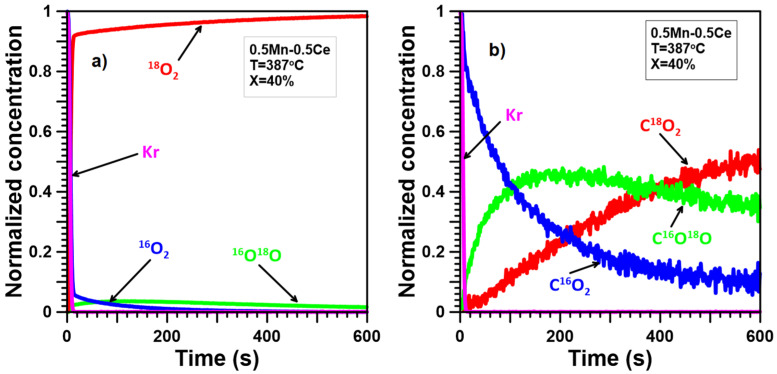
ITKA results obtained for the 0.5Mn-0.5Ce catalyst at the temperature corresponding to 40% soot conversion ((**a**)—changes in ^16^O_2_, ^16^O^18^O, and ^18^O_2_ concentration; (**b**)—changes in C^16^O_2_, C^16^O^18^O, and C^18^O_2_ concentration observed after switching from ^16^O_2_/Kr/Ar/He to ^18^O_2_/Ar/He).

**Figure 8 molecules-30-00343-f008:**
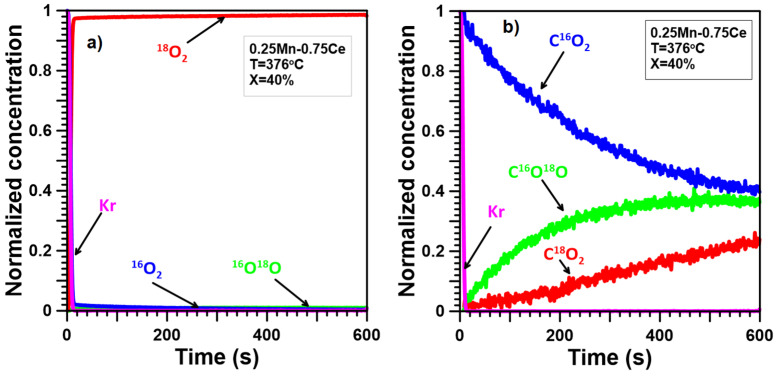
ITKA results obtained for the 0.25Mn-0.75Ce catalyst at the temperature corresponding to 40% soot conversion ((**a**)—changes in ^16^O_2_, ^16^O^18^O, and ^18^O_2_ concentration; (**b**)—changes in C^16^O_2_, C^16^O^18^O, and C^18^O_2_ concentration observed after switching from ^16^O_2_/Kr/Ar/He to ^18^O_2_/Ar/He).

**Figure 9 molecules-30-00343-f009:**
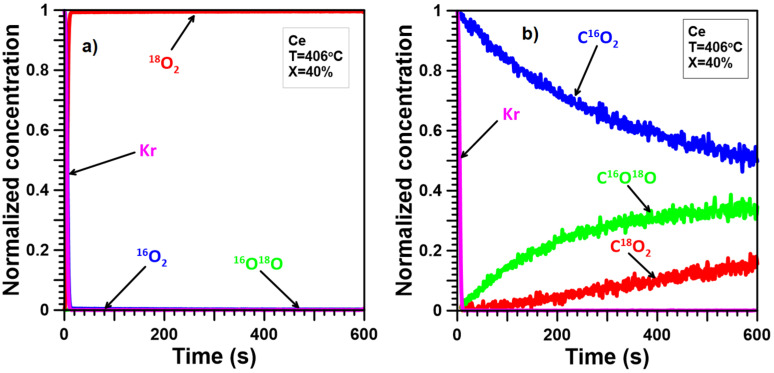
ITKA results obtained for the Ce catalyst at the temperature corresponding to 40% soot conversion ((**a**)—changes in ^16^O_2_, ^16^O^18^O, and ^18^O_2_ concentration; (**b**)—changes in C^16^O_2_, C^16^O^18^O, and C^18^O_2_ concentration observed after switching from ^16^O_2_/Kr/Ar/He to ^18^O_2_/Ar/He).

**Table 1 molecules-30-00343-t001:** Metal content in manganese, manganese–cerium, and cerium catalysts determined via XRF.

Catalyst	Metal Content (% wt.)		Molar Ratio (Mn:Ce)
	Mn	Ce	
Mn	71.0	0.0	1.00:0.00
0.75Mn-0.25Ce	40.2	31.7	0.76:0.24
0.5Mn-0.5Ce	22.6	53.1	0.52:0.48
0.25Mn-0.75Ce	9.6	70.6	0.26:0.74
Ce	0.0	81.2	0.00:1.00

**Table 2 molecules-30-00343-t002:** Total BET surface area, average pore diameter, and volume of pores determined for manganese, manganese–cerium, and cerium catalysts.

Catalyst	Total BET Surface Area (m^2^/g)	Average Pore Diameter (nm)	Volume of Pores (cm³/g)
Mn	7.5	30.94	0.05
0.75Mn-0.25Ce	76.5	7.77	0.14
0.5Mn-0.5Ce	101.6	5.81	0.10
0.25Mn-0.75Ce	102.0	6.11	0.07
Ce	85.9	5.51	0.04

**Table 3 molecules-30-00343-t003:** Phase composition and crystallite size obtained via XRD for manganese, manganese–cerium, and cerium catalysts.

Catalyst	Crystallographic Phases			Mean Size of Crystallites (nm)
	Chemical Formula	Content (%)	Crystallite Size (nm)	
Mn	Mn_3_O_4_	95	70.8	69.7
	Mn_2_O_3_	5	49.8	
0.75Mn-0.25Ce	Mn_3_O_4_	34	29.8	21.9
	CeO_2_	66	17.8	
0.5Mn-0.5Ce	Mn_3_O_4_	8	18.3	11.5
	CeO_2_	92	10.9	
0.25Mn-0.75Ce	Mn_3_O_4_	2	7.2	13.1
	CeO_2_	98	13.3	
Ce	CeO_2_	100	16.4	16.4

## Data Availability

Data are contained within the Article and Appendix A.

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
