# Peer review of "Isotopic Transient Kinetic Analysis of Soot Oxidation on Mn3O4, Mn3O4-CeO2, and CeO2 Catalysts in Tight Contact Conditions"

_molecules, 2025, doi:10.3390/molecules30020343_

Round 1
Reviewer 1 Report
Comments and Suggestions for Authors
The work on reaction mechanism of CeO2 and MnOx-based catalysts towards soot oxidation was very important. Isotopic transient kinetic analysis of soot oxidation has been proved to be an efficient route. It was a very in-depth work, but rather incomplete considering publication in Catalysts. Some suggestions were listed as follows.
1. The isotopic analysis in H2-TPR should be done, along with XPS and TPO results, to determine the dedication of surface oxygen and lattice oxygen on soot oxidation process. In Line 238, "Of course, these results cannot completely exclude the participation of chemisorbed oxygen and/or lattice oxygen only from the surface in the soot oxidation process, but its influence may only be very small". This statement lacks sufficient evidence.
2. In the Mn-Ce catalysts, how to understand the effects of different valence of Mn on the kinetics of soot oxidation? The interaction between Ce and Mn on the oxygen defects should be studied in depth.
3. Recycled TPO can be carried out in view of the practical applications.
Author Response
Comments: The work on reaction mechanism of CeO2 and MnOx-based catalysts towards soot oxidation was very important. Isotopic transient kinetic analysis of soot oxidation has been proved to be an efficient route. It was a very in-depth work, but rather incomplete considering publication in Catalysts. Some suggestions were listed as follows.
Response: Thank you very much for taking the time to review this manuscript. Please find the detailed responses below:
Comments 1: The isotopic analysis in H2-TPR should be done, along with XPS and TPO results, to determine the dedication of surface oxygen and lattice oxygen on the soot oxidation process. In Line 238, "Of course, these results cannot completely exclude the participation of chemisorbed oxygen and/or lattice oxygen only from the surface in the soot oxidation process, but its influence may only be very small". This statement lacks sufficient evidence.
Response 1: The ITKA results (Figure 9b) clearly indicate the Mars-van Krevelen reaction mechanism - the participation of oxygen from the crystal lattice-bulk of the catalyst. However, during the soot oxidation reaction, chemisorbed oxygen can be present on the catalyst surface, therefore its participation in the soot oxidation cannot be ruled out. So, we added this sentence - indicated by the Reviewer.
In this point, it should be emphasized that all the indicated methods (TPR, TPO, XPS) show us the catalyst surface in conditions significantly different from the real reaction conditions, only the ITKA results come from the real reaction conditions. So, if we see something on the catalyst surface before a reaction, it may be completely different during the reaction.
How to do the isotopic analysis in H2-TPR? Such an analysis would only make sense if it were possible to replace only the chemically adsorbed oxygen and/or the oxygen from the first crystal layer of the catalyst. When we replace all oxygen atoms in the catalyst sample we obtain the same results as during the normal H2-TPR measurement. The second problem with the isotopic analysis in H2-TPR concerns the analysis of small amounts of isotopically-labeled and non-labeled water molecules, which can be strongly adsorbed on the elements of apparatuses.
Comments 2: In the Mn-Ce catalysts, how to understand the effects of different valence of Mn on the kinetics of soot oxidation? The interaction between Ce and Mn on the oxygen defects should be studied in depth.
Response 2: All Mn-Ce catalysts have practically the same phase composition (both the surface layer and the whole bulk) which was confirmed by XRD, XPS, and Raman spectroscopy as well as TPR and TPO results. On the other hand, large amounts of Mn3O4 (min. 25%) and CeO2 (min. 25%) and their very good mutual mixing make it impossible to observe effects such as the entry of atoms of a given element, e.g. Mn into the crystal lattice of oxide, e.g. CeO2. Such an effect could be observed if the Mn concentration was a few percent. Here, all applied methods show only signals from the main crystallographic phases, other possible signals were obscured/covered, so is difficult to add something else to this discussion. This has already been explained, among others, in lines 103-105.
Comments 3: Recycled TPO can be carried out in view of the practical applications.
Response 3: I'm not entirely sure what the Reviewer meant by this question. Is it possible to conduct TPO tests on used catalyst samples? or; Is it possible to conduct activity tests on used catalyst samples? or; something others? It does not make sense to carry out TPO measurements on used catalyst samples because their surface is oxidized. The re-run of activity tests on samples of used catalysts would require a change in sample preparation methodology because the catalyst-soot sample after pressed was diluted with sand (separation of the catalyst sample from sand is impossible). We cannot change the catalyst sample preparation methodology because it would give different results. At this point, we would like to emphasize that the main purpose of this publication was to analyze the reaction mechanism and not the catalyst lifetime. There are results in the literature suggesting the potential usefulness of such materials in real applications, but the problem of explaining the reaction mechanism still exists.
Reviewer 2 Report
Comments and Suggestions for Authors
This work investigated the reaction mechanism of Mn-Ce catalyst via isotopic transient kinetic analysis. The Mn3O4, CeO2, and Mn-Ce with different Mn/Ce radio catalysts were prepared and the catalytic activity was tested. A series of characterization methods and isotopic transient kinetic analysis were conducted to research the reaction mechanism. But this manuscript could be better with the improvement in language presentation.
Question 1:
Page 4, the peaks of Mn 2p3/2 is too many and are unreasonable.
Question 2:
It is suggested to conduct the repeat experiments of activity test to make sure your results reliable.
Question 3:
Page 6, Line 156, the hydrogen consumption should be listed to prove the conjecture here.
Question 44:
Why the catalytic activity of Ce catalytic is better than Mn catalytic?
Question 5:
Page 6, Line 169, how could the author come to this conclusion?
Comments on the Quality of English LanguageStatements should be easy to understand and clearly express the author's meaning.
Author Response
Comments: This work investigated the reaction mechanism of Mn-Ce catalyst via isotopic transient kinetic analysis. The Mn3O4, CeO2, and Mn-Ce with different Mn/Ce radio catalysts were prepared and the catalytic activity was tested. A series of characterization methods and isotopic transient kinetic analysis were conducted to research the reaction mechanism. But this manuscript could be better with the improvement in language presentation.
Response: Thank you very much for taking the time to review this manuscript. Below are detailed answers to your questions. The expert also checked the language of the publication.
Question 1:
Page 4, the peaks of Mn 2p3/2 is too many and are unreasonable.
Response 1: We partially agree with your opinion. However, this analysis is very complicated because there are small differences between the individual manganese oxides. Therefore, the analysis of Mn2p3/2 was performed by mathematically fitting the model data for Mn3O4 (all possible signals) and excluding other oxides by obtaining a poorer fit. To exclude other manganese oxides, we use, among others, data from the paper: C. Biesinger, B.P. Payne, A.P. Grosvenor, L.W.M. Lau, A.R. Gerson, R.S.C. Smart, Appl. Surf. Sci. 257, (2011) 2717-2730.
Question 2:
It is suggested to conduct the repeat experiments of activity test to make sure your results reliable.
Response 2: The example was attached in the supplementary materials file. The obtained results show good repeatability of activity tests.
Question 3:
Page 6, Line 156, the hydrogen consumption should be listed to prove the conjecture here.
Response 3: The hydrogen consumption was calculated and attached in the supplementary materials file.
Question 4:
Why the catalytic activity of Ce catalytic is better than Mn catalytic?
Response 4: This is a very general question. These are two completely different elements, with different electron configurations, different properties, etc. The main goal of this study was to analyze the reaction mechanism and this goal was achieved. The mechanism of soot oxidation on the Ce catalyst is different from that on the Mn catalyst. This can be one of the main causes.
Question 5:
Page 6, Line 169, how could the author come to this conclusion?
Response 5: We have changed part of the sentence (page 6, line 169) because its meaning has probably been interpreted differently. The new sentence: “It means that oxygen (removed during reduction at 500 °C) in the crystal lattice of the Mn-Ce catalysts can be almost completely replenished at temperatures below the temperatures of soot oxidation (Figure 3)”
Round 2
Reviewer 1 Report
Comments and Suggestions for Authors
The paper can be accepted in present form.
Author Response
Thank you very much for taking the time to review this manuscript and for your positive decision.